# Reliable Condensation Curing Silicone Elastomers with Tailorable Properties

**DOI:** 10.3390/molecules26010082

**Published:** 2020-12-27

**Authors:** Alena Jurásková, Stefan Møller Olsen, Kim Dam-Johansen, Michael A. Brook, Anne Ladegaard Skov

**Affiliations:** 1DPC, Department of Chemical and Biochemical Engineering, Technical University of Denmark (DTU), Building 227, 2800 Kgs. Lyngby, Denmark; alejur@kt.dtu.dk; 2Hempel A/S, Lundtoftegårdsvej 91, 2800 Kgs. Lyngby, Denmark; STMO@hempel.com; 3CoaST, Department of Chemical and Biochemical Engineering, Technical University of Denmark (DTU), Building 229, 2800 Kgs. Lyngby, Denmark; kdj@kt.dtu.dk; 4Chemistry and Chemical Biology, McMaster University, 1280 Main St. W, Hamilton, ON L8S 4M1, Canada; mabrook@mcmaster.ca

**Keywords:** silicone elastomers, stability, network structure, condensation curing, reinforcing cross-linker domains, coatings

## Abstract

The long-term stability of condensation curing silicone elastomers can be affected by many factors such as curing environment, cross-linker type and concentration, and catalyst concentration. Mechanically unstable silicone elastomers may lead to undesirable application failure or reduced lifetime. This study investigates the stability of different condensation curing silicone elastomer compositions. Elastomers are prepared via the reaction of telechelic silanol-terminated polydimethylsiloxane (HO-PDMS-OH) with trimethoxysilane-terminated polysiloxane ((MeO)_3_Si-PDMS-Si(OMe)_3_) and ethoxy-terminated octakis(dimethylsiloxy)-T8-silsesquioxane ((QM^OEt^)_8_), respectively. Two post-curing reactions are found to significantly affect both the stability of mechanical properties over time and final properties of the resulting elastomers: Namely, the condensation of dangling and/or unreacted polymer chains, and the reaction between cross-linker molecules. Findings from the stability study are then used to prepare reliable silicone elastomer coatings. Coating properties are tailored by varying the cross-linker molecular weight, type, and concentration. Finally, it is shown that, by proper choice of all three parameters, a coating with excellent scratch resistance and electrical breakdown strength can be produced even without an addition of fillers.

## 1. Introduction

Condensation curing silicone elastomers are commonly used as protective coatings and sealants [1,2,3,4,5]. They are prepared via a condensation reaction between a HO-PDMS-OH and a silane cross-linker with hydrolysable groups such as amino, amide, acyloxy, ketoxime, or alkoxy [6]. Alkoxysilanes are preferred for this reaction, as they are non-corrosive, inexpensive, and do not produce toxic by-products during curing [7]. The condensation reaction allows efficient curing in ambient environment, which represents a significant advantage over silicone elastomers prepared by addition, radical, or UV curing. Besides the mild curing conditions, condensation curing silicone elastomer applications also benefit from the superior properties common to silicone elastomers, such as high flexibility, low surface energy, high chemical resistance, and excellent thermal stability [1,8,9]. Nevertheless, one drawback of condensation curing silicone elastomers is poor control over the curing reaction [6,9,10].

In our previous study [10], we showed that an inappropriate choice of the network formulation significantly compromises long-term elastomer stability such that the elastomer’s properties change continuously over time (the study was terminated after 6 months, at which time changes were still being observed). Several factors, such as cross-linker volatility and purity, as well as catalyst concentration, were shown to significantly affect network formation and thus also contribute to poor reliability and reproducibility of silicone elastomers. For example, low molecular weight alkoxysilane cross-linkers tend to evaporate from the elastomer mixture and are prone to premature hydrolysis-condensation reactions (Figure 1a,b) [10,11,12,13,14]. In addition, HO-PDMS-OH condenses in the presence of a tin catalyst, which further contributes to network formulation difficulties due to the loss of reactive groups participating in the crosslinking reaction (Figure 1c) [10,15]. The cross-linker volatility and condensation of HO-PDMS-OH affect the true stoichiometric ratio between the functional group of cross-linker and polymer: r = f[cross-linker]/2[HO-PDMS-OH, where […] denotes the original concentration of the species. Depending on the final *r*, we hypothesized two types of post-curing effect leading to long-term elastomer instability: The reaction between cross-linker molecules in the case of cross-linker excess, and the condensation of dangling and/or unreacted silanol-terminated polymer chains in the case of cross-linker deficit [10].

It was obvious that in order to develop stable condensation curing silicone elastomers, whose properties do not change significantly over time, potential post-curing effects must be investigated. The first part of this study is therefore focused on optimizing elastomer formulations, paying particular attention to the relationship between the stoichiometric ratio and the type/extent of the post-curing reaction. Trimethoxysilane-terminated polysiloxanes ((MeO)_3_Si-PDMS-Si(OMe)_3_) and ethoxy-terminated silsesquioxane ((QM^OEt^)_8_) are used as cross-linkers. The choice of the cross-linkers is based on previous research [10] in which we showed that low molecular weight alkoxysilane cross-linkers constitute the largest obstacle towards developing stable silicone elastomers with reproducible material performance. This is because of their volatility, which causes the curing to become highly dependent on both sample dimensions and the surrounding environment. In addition, we also showed that methyltrimethoxysilane does not allow curing of thin films (≤200 µm), unless the films are covered during the initial stage of the curing reaction. For thin silicone elastomer films, which need to be cured in open air, higher molecular weight alkoxysiloxane cross-linkers should therefore be used.

After the thorough study of the relation between condensation curing silicone elastomer formulation and elastomer stability, the post-curing effects were well understood. The second part of this work therefore focuses on the performance of stable elastomer films and the possibility to tailor their properties, in particular Young’s modulus, elongation at break, electrical breakdown strength, and scratch resistance.

## 2. Results and Discussion

### 2.1. Optimization of Condensation Curing Silicone Elastomer Formulations

In our previous study, we hypothesized two post-curing effects leading to condensation curing silicone elastomer instability: The condensation of dangling and/or unreacted silanol-terminated polymer chains, and the reaction between cross-linker molecules [10]. In order to design stable condensation curing silicone elastomers, a better understanding of these post-curing reactions is needed. The work presented in this chapter is therefore focused on the mechanical stability of different condensation curing silicone elastomer formulations. Non-volatile alkoxy-terminated polysiloxanes and silsesquioxane cross-linkers were used to eliminate the negative effects of the volatile alkoxysilane cross-linkers. All elastomers were prepared using the same dibutyltin-dilaurate (Sn_DL) concentration (0.5 wt%) and film thickness (~100 µm). The individual silicone elastomer compositions are summarized in Table A2.

In the ideal case, an optimum network, so-called a model network, will be obtained using a stoichiometry between functional groups of cross-linker and polymer (*r* = 1) [9]. In such network, all functional groups will be reacted, as illustrated in Figure 2. Thereby, any eventual post-curing reactions causing network instability will be hindered. However, due to the numerous side reactions that take place during condensation curing and the steric hindrance of functional groups, such a model network is never actually formed. Hence, in order to prepare well-characterized and stable silicone elastomers, a stoichiometric ratio optimization study must be conducted for each individual cross-linker.

#### 2.1.1. Trimethoxysilane-Terminated Polysiloxane Cross-Linkers

The stability of elastomer films (~100 µm) consisting of C2T and Di-10, Di-50, and Di-400, respectively, was evaluated via changes in Young’s modulus over time. To investigate the impact of functional group imbalance on post-curing reactions, the stoichiometric ratio of these films was varied from *r* = 1.5 to *r* = 20 (Table A2). Figure 3 shows that E_C2T+Di-10 and E_C2T+Di-50 displayed similar trends with respect to alteration of the Young´s modulus over time. In particular, low stoichiometric ratio films (*r* = 1.5 and 2) showed a three- to four-fold increase in Young´s modulus over the course of four months. High stoichiometric ratio films (*r* = 15 and 20), on the other hand, exhibited a steep increase in Young´s modulus during the first three weeks, after which it reached a stable value. Finally, films with stoichiometric ratios of *r* = 5 and *r* = 10 displayed the smallest change in Young´s modulus over time. However, E-C2T+Di-400 showed a different trend, in which the Young´s modulus increased during first 30 days independently of the stoichiometric ratio, after which it reached a stable value in films with *r* ≥ 2.

To better understand the processes behind the increase in Young´s modulus over time, ^1^H-NMR and SEC analysis of extracts from the elastomer films were conducted and the results are summarized in Figure 4. The amount of unreacted PDMS over time was calculated from ^1^H-NMR spectra (Figure A2). Elastomers E_C2T+Di-50 with *r* = 5, 10, and 20 contained between 2 to 4 wt% of extractable PDMS. The corresponding SEC eluograms of the extracts showed a double peak at retention volumes between 17 and 22 mL. This double peak, whose intensity did not decrease over time, was also found in the eluograms of the C2T and Silmer cross-linkers (Figure A1). The extractable PDMS eluting at the retention volume 17–22 mL is therefore assumed to be a non-functional, low molecular weight PDMS originating from the manufacturing of the polymers/cross-linkers. Elastomer E_C2T+Di-50_r1.5 contained a significant amount of extractable PDMS, which decreased over time. The SEC eluograms showed, apart from the double peak at 17–22 mL, a peak at retention volumes between 11 and 17 mL, which corresponds to the retention volume of unreacted HO-PDMS-OH (C2T). The amount of PDMS extractable from E_C2T+Di-400 decreased over the first 30 days regardless of *r*. The peak at retention volumes between 11 and 17 mL can be then attributed to unreacted C2T and Silmer Di-400. After 27 days, extracts from E_C2T+Di-400 with *r* = 5, 10, and 20 reached stable values of ~5 wt% as a natural result of non-functional PDMS residues from the starting material.

Combining the knowledge gained from the development of Young´s modulus over time (Figure 3) with the extract analysis (Figure 4), several conclusions can be drawn regarding the post-curing reactions. First, the post-curing reaction between unreacted and/or dangling polymer chains is generally a slow process, lasting several months. As expected, this reaction takes place at lower stoichiometric ratios, where more dangling substructures are present. For the formulations studied here, this post-curing reaction occurred for *r* < 5 when Di-10 and Di-50 were used as cross-linkers, and occurred to a minor extent for all tested stoichiometric ratios when Di-400 was used as the cross-linker. This is due to the high molecular weight of Di-400, which is comparable to that of C2T. Second, the post-curing reaction between cross-linker molecules is a comparatively fast process that is completed within approximately 3 weeks, as evidenced by the rapid increase in Young´s modulus over time and the 0 wt% of extractable HO-PDMS-OH in the elastomer network. In the formulations tested here, the reaction between cross-linker molecules become significant at *r* > 10. As demonstrated later on in this work, the right choice of cross-linker leads to a favorable formation of cross-linker domains that provide reinforcement to the elastomer without the addition of fillers. Third, post-curing effects were smallest at 2 < *r* < 10 as elastomers prepared within this range contain minimal amounts of both unreacted/dangling polymer chains and cross-linker domains. This is recognized by the combination of the minimum change in Young´s modulus over time and 0 wt% of extractable HO-PDMS-OH. The fact that the stoichiometric ratio, which produces the fewest post-curing reactions, is relatively high—far above the stoichiometric balance of *r* = 1—appears counterintuitive, but may be due to following factors: (1) Steric hindrance of the –OCH_3_ groups of (MeO)_3_Si-PDMS-Si(OMe)_3_ cross-linker may hinder the reaction of all cross-linker functional groups; (2) the cross-linker molecules may undergo a condensation reaction during the main curing process, thereby creating unavoidable cross-linker domains, which decrease the true stoichiometry of the reaction mixture. A schematic illustration of the above-described effects of stoichiometric ratio and elastomer age on elastomer network structure and stability is summarized in Figure 5.

#### 2.1.2. Silsesquioxane Cross-Linker

A formulation optimization study similar to that presented for trimethoxysilane-terminated polysiloxanes cross-linkers was also conducted for (QM^OEt^)_8_. The silsesquioxane cross-linker benefited from the fact that the Si(CH_3_)_2_ signal of the (QM^OEt^)_8_ cross-linker is distinguishable from the Si(CH_3_)_2_ signal of the PDMS (C2T) (Figure 6), enabling an even more comprehensive investigation of the condensation curing process.

The amounts of unreacted PDMS and (QM^OEt^)_8_ over time were calculated from ^1^H-NMR spectra (Figure A3), and the results are summarized in Figure 7. Silicone elastomers with *r* = 0.5 and 1 contained 0 wt% of unreacted (QM^OEt^)_8_ and a significant amount of extractable PDMS, which decreased over time due to the slow condensation of HO-PDMS-OH. Silicone elastomers with *r* = 3 and 5 contained close to 0 wt% of unreacted (QM^OEt^)_8_ and only ~4 wt% of extractable, non-functional low molecular weight PDMS originating from the manufacturing of the C2T polymer (Figure A1). Elastomers with a stoichiometric ratio in the interval 3 ≤ *r* ≤ 5 are thus the most stable elastomers, experiencing the fewest post-curing effects. Again, this ratio is far above the expected optimal stoichiometric balance (*r* = 1), suggesting the formation of cross-linker domains during the main curing reaction, as in the previously studied formulations. Silicone elastomers with *r* = 8 and 15 contained 3 and 6.5 wt% of unreacted (QM^OEt^)_8_ molecules, respectively. The concentration of the unreacted cross-linker decreased over time, reaching 0 wt% after 17 days as additional cross-linker domains were created. Since silsesquioxanes are generally known to be self-reinforcing cross-linkers [16,17,18,19,20,21], (QM^OEt^)_8_ cross-linker domains are expected to have a positive effect on elastomer film strength.

### 2.2. Mechanically Stable Silicone Elastomer Films and Their Properties

Findings from the formulation optimization study presented above were used to design stable condensation curing silicone elastomer films. With both possible post-curing reactions in mind, different network structures were prepared by changing the cross-linker type (polysiloxane or silsesquioxane), cross-linker chain length (Di-10, 50, or 200), or stochiometric ratio (*r* = 3, 5 or 15). The individual silicone elastomer compositions are summarized in Table A3. While all films were cured within a few hours, the measurements presented in this chapter were conducted after storage in a climate chamber for 27 days, the time required to achieve complete cross-linker domains creation (see Section 2.1). After 27 days, all films, except for commercial coating E_Ref, contained low amounts of sol fraction ranging from 3 to 5 wt% (Table A4), which can be assigned to the non-reactive PDMS created during polymer/cross-linker synthesis (Figure 4 and Figure A1). The sol fraction of E_Ref was found to be 6–7 wt%. Apart from the double peak at retention volumes between 17 and 22 mL, SEC analysis also showed a peak at retention volumes between 11 and 17 mL (Figure A4). This second peak can be assigned to a silicone oil, either added as a plasticizer or originating from non-reacted HO-PDMS-OH.

Network elasticity and rigidity were tailored via cross-linker domain concentration and density. Figure 8 and Figure 9 show that, as expected, Young´s modulus was increasing, and elongation at break was decreasing with increasing *r.* These changes were even more pronounced when using Di-10 and (QM^OEt^)_8_, as the cross-linker domains become more dense. In contrast, the smallest change in both Young´s modulus and elongation at break with increasing *r* was found in films cross-linked by Di-400 due to its high molecular weight, which is comparable to that of C2T (Table 1). It should be noted that, despite the relatively low Young´s modulus of elastomer films E_C2T+Di-400_r5, E_C2T+Di-400_r15, and E_Di-400 (0.61, 0.67 MPa, and 0.7 MPa, respectively), the sol fraction of each remained below 5 wt% (Table A4). Additionally, the elastomer E_C2T+(QM^OEt^)_8__r15 exhibited opacity when elongated. For better understanding of this behavior, which was not observed for any of the other elastomers, the elastomer E_C2T+(QM^OEt^)_8__r15 was investigated using SEM in both unstretched and stretched state. The sample preparation procedure can be found in Figure A5. As it can be seen in Figure 10, the elastomer undergoes a significant surface change when stretched, which explains the observed opacity and is believed to be a result of (QM^OEt^)_8_ domains aggregation and crystallization.

Electrical breakdown (EBD) strength is an indicator of film homogeneity, since film imperfections lead to inhomogeneous fields and thus premature film failure [22]. Increased Young’s modulus leads to increased electrical breakdown strength, but only if the film is homogenous [23,24]. The presence of cross-linker domains was found to positively affect electrical breakdown strength. The highest electric breakdown strength of 130 µm/V was obtained for elastomer E_C2T+(QM^OEt^)_8__r15 (Figure 11). This is approximately 30% higher than the electrical breakdown strength of the reference coating (E_Ref), which is a commercial condensation curing silicone coating containing reinforcing fillers. It is also significantly higher than values reported for addition curing silicone elastomers, such as Sylgard, Ecoflex, and Elastosil [25].

Scratch resistance, together with coating/substrate adhesion, is one of the most important parameters for materials used as protective coatings. A poor scratch resistance and/or adhesion to the substrate can lead to reduction of coating performance, reliability, and lifetime [1,26,27,28]. In this study, all the tested elastomers, independently of their composition, showed initial cohesive failure before failing adhesively when scratched, signifying a good adhesion to their substrate, namely Hempel’s Nexus II 27400 [26,27]. The scratch resistance was then found to improve significantly with increasing *r* for elastomers cross-linked by Di-10 and (QM^OEt^)_8_ (Figure 12). Elastomers E_C2T+Di-10_r15 and E_C2T+(QM^OEt^)_8__r15 displayed a scratch resistance comparable to that of the reference coating, E_Ref, which, unlike the elastomers investigated here, contains reinforcing fillers. While this finding once again shows the positive effect of Di-10 and (QM^OEt^)_8_ cross-linker domains, a negative correlation between increasing *r* and scratch resistance was observed for elastomers cross-linked by Di-50. This can be explained by the high weight percentage of Di-50 cross-linker in the elastomer (Table A3), which caused the elastomer E_C2T+Di-50_r15 (~58 wt% of the Di-50) to lose its elasticity. Zero or negligible difference in scratch resistance with increasing *r* was reported for E_C2T+Di-400 and E_Di-400, as the high molecular weight of the Di-400 cross-linker does not allow the formation of strongly reinforcing domains due to the long distance between cross-links. Noticeably, even though E_C2T+(QM^OEt^)_8__15 showed excellent “single” scratch resistance, its “multiple” scratch resistance was significantly lower. On the other hand, E_C2T+Di-10_r15 displayed excellent “single” and “multiple” scratch resistance. Both elastomers (E_C2T+(QM^OEt^)_8__15 and E_C2T+Di-10_r15) contain similar weight percentages of cross-linker (20 and 23 wt%, respectively), suggesting that the reduced “multiple” scratch resistance displayed by E_C2T+(QM^OEt^)_8__15 can be attributed to the rigidity of the (QM^OEt^)_8_ domains. The more elastic Di-10 domains performed well in both “single” and “multiple” scratch tests, indicating that not only cross-linker domains size, but also rigidity/elasticity, is important for scratch resistance. For better understanding of the difference in the network structure containing Di-10, Di-50, Di-400, and (QM^OEt^)_8_ domains, respectively, Figure 13 introduces a simplified cartoon of network structures prepared at high *r*.

The results above demonstrate that introducing cross-linker domains with specific structural properties can significantly improve silicone elastomer performance. For example, silicone elastomers E_C2T+(QM^OEt^)_8__15 and E_C2T+Di-10_r15 were shown to match, or even outperform, commercial coating E_Ref containing fillers and other additives in both electrical breakdown strength and scratch resistance. The addition of fillers is commonly used to improve the mechanical properties of silicone elastomers. However, it often leads to an undesirable Mullins effect, a loss of coating transparency, and an increase in Young´s modulus [29]. Creating cross-linker domains in the elastomer network represents an alternative method for improving elastomers’ mechanical properties without introducing the Mullins effect or compromising elastomer transparency. Although the Young´s modulus of elastomers E_C2T+(QM^OEt^)_8__15 and E_C2T+Di-10_r15 is significantly higher than that of commercial coating E_Ref, it should be noted that the latter contains a silicone oil in the sol fraction, which lowers its Young´s modulus (Figure A4).

## 3. Materials and Methods

### 3.1. Materials

Dibutyltin-dilaurate catalyst (Sn_DL)—Tib kat 218 (produced by TIB Chemicals) and Hempel´s Nexus II 27400 was kindly provided by Hempel. Linear, trimethoxysilane-terminated polysiloxanes ((MeO)_3_Si-PDMS-Si(OMe)_3_)—Silmer TMS Di-10, Silmer TMS Di-50, and Silmer TMS Di-400), were purchased from Siltech. The molecular weights of (MeO)_3_Si-PDMS-Si(OMe)_3_ were determined by SEC and ^1^H NMR (Table 1 and Table A1). The names of the polysiloxane cross-linkers are shortened to Di-10, Di-50, and Di-400, respectively, throughout the manuscript. Silanol-terminated polydimethylsiloxane (HO-PDMS-OH; C2T) was obtained from Wacker Chemie. The molecular weight of C2T was determined by SEC (Table 1). Ethoxy-terminated octakis(dimethylsiloxy)-T8-silsesquioxane ((QM^OEt^)_8_) was synthetized via ethanolysis of octakis(dimethylsiloxy)-T8-silsesquioxane ((QM^H^)_8_) in the presence of Karstedt´s catalyst [30]. Hostaphan RN 190/190 µm (polyester plastic carrier) was supplied by Mitsubishi Polyester Film. Syringe filter (diameter: 25 mm, pore size: 0.45, PTFE membrane) was purchased from Fisher Scientific.

### 3.2. Elastomer Film Preparation

Example of the preparation procedure for elastomer E_C2T+Di-10_r3: Silanol-terminated polydimethylsiloxane—C2T and trimethoxysilane-terminated polysiloxane—Di-10 were mixed in a mixing container using a FlackTech speed mixer at 2700 rpm for 3 min. The stoichiometric ratio between cross-linker and polymer functional groups (*r* = 6[(MeO)_3_Si-PDMS-Si(OMe)_3_]/2[HO-PDSM-OH]) was *r* = 3. Subsequently, 0.5 wt% of Sn_DL was added before mixing in a FlackTech speed mixer at 2700 rpm for another 3 min. After a homogeneous mixture was obtained, the sample was coated into a thin film by a coating knife with a nominal coating height of 200 µm onto a polyester carrier. For the scratch resistance measurements, the mixture was coated onto a tie-coat—Hempel´s Nexus II 27400, which was previously coated onto a polyester carrier using a coating knife with a nominal coating height of 200 µm. The films were subsequently cured in a climate chamber at 25 °C and 80% relative humidity.

The elastomer compositions for the long-term stability study can be found in Table A2.

The elastomer compositions for tensile, electrical breakdown strength, and scratch resistance measurements can be found in Table A3.

The abbreviation, molecular weight, and chemical structure of the compounds used in the elastomer formulations are summarized in Table 1.

### 3.3. Evaluation of the Elastomer Film Stability

The silicone elastomer films, for which composition details can be found in Table A2, were used to study the effect of elastomer composition on network structure and stability. The following methods were used to characterize film stability: **Mechanical stability** (Figure 14a) was evaluated by tracking changes in Young´s modulus over time. The Young´s modulus was determined from the tangent of the linear region of the stress–strain curves at low strains (up to approximately 20% strain). Standard deviations were calculated from five tensile measurements for each sample composition. The tensile measurements were performed on an ARES-G2 rheometer using SER geometry according to a previously reported procedure [10]. **Chemical composition of the extract and wt% of extractable PDMS/cross-linker** (Figure 14b) was determined using Proton Nuclear Magnetic Resonance (^1^H NMR). A specimen 25 mm in length and 6 mm in width was cut from the silicone elastomer film (~100 µm), weighed, and extracted in 1 mL of chloroform-*d* for 24 h. After the extraction, a known amount of naphthalene was added, and the solution was transported into a NMR tube. ^1^H NMR characterization was performed on a 7 Tesla Spectrospin-Bruker AC 300MHz spectrometer at room temperature. ^1^H NMR spectra were analyzed using MestReNova. Detailed calculation of the wt% of extractable PDMS can be found in Figure A2 and Figure A3. **Molecular weight of extractable PDMS** (Figure 14c) was obtained using Size-Exclusion Chromatography (SEC). Two specimens, 25 mm in length and 6 mm in width, were cut from the silicone elastomer film (~100 µm), weighed, and extracted in 1.2 mL of toluene for 24 h. The extract was filtered through a syringe filter. SEC was performed on a TOSOH EcoSEC HLC-8320GPC system equipped with an EcoSEC RI detector. This system was fitted with two SDV LINEAR S 5 µm 8 × 300 mm columns in series, protected by a GUARD column (SDV 5 µm 8 × 50 mm), all supplied by PSS. Samples were run in toluene at 35 °C. Molar mass characteristics were calculated using PSS WinGPC Unity, Build 9350 software, and linear PDMS standards acquired from PSS.

### 3.4. Evaluation of the Silicone Elastomer Films Performance

The silicone elastomer film compositions can be found in Table A3. The following properties were tested in order to evaluate elastomer performance: **Sol fraction** was determined using both ^1^H NMR (sol fraction consisting purely from the extractable PDMS species) and sample weight lost (sol (%) = (m_i_ − m_d_)/m_i_ × 100, where m_i_ is the original sample weight and m_d_ is the sample weight after the extraction and drying). For more details on the extraction, see the procedure in Section 3.3. The 24-h extraction time was found to be sufficient, as no additional weight loss was observed after a second extraction of the same specimen. **Tensile properties** were performed on an ARES-G2 rheometer using SER geometry according to a previously described procedure [10]. **Scratch resistance** was measured using a Motorized Clemen Scratch Tester equipped with a Ø 1mm ball tool. Film destruction was observed visually. Two different parameters were evaluated—“single” scratch resistance and “multiple” scratch resistance. The “single” scratch resistance was determined as the maximum load at which the coating continues to resist penetration. The “multiple” scratch resistance was determined as the maximum load at which the coating remained unpenetrated after three consecutive scratches at the same place. Both “single” and “multiple” scratch procedures were repeated three times for each coating composition. If the coating was penetrated during one or more of the three repetitions, the load was lowered and the whole procedure was repeated. **Electrical breakdown strength** was measured using an instrument built in-house following the international standards (IEC 60243-1 (1998) and IEC 60243-2 (2001)) [31,32]. A silicone elastomer film approximately 100 µm thick was placed on a plastic frame and positioned inside the breakdown instrument between two semi-spherical stainless-steel electrodes. A voltage ramp of 100 V/s was then applied until the elastomer short-circuited. The electrical breakdown strength was calculated as the voltage at breakdown divided by sample thickness. The standard deviation was calculated from 10 breakdown measurements for each elastomer film. **Scanning electron microscope (SEM)**—Inspect S, FEI company—was used to evaluate the changes in the elastomer E_C2T+(QM^OEt^)_8__15 before and after stretch. Images were taken in a low vacuum mode. The sample preparation can be found in Figure A5.

The exact sample thickness for the tensile, scratch resistance, and electrical breakdown strength measurements was determined using an optical microscope according to a previously described procedure [10].

## 4. Conclusions

In this study, post-curing reactions occurring in condensation curing silicone elastomers were thoroughly investigated using rheology, SEC, and ^1^H NMR, findings from which were correlated to the elastomers’ network structure. Depending on the stoichiometric ratio, two types of post-curing reaction were shown to occur: Reactions between unreacted and/or dangling polymer chains, and reactions between cross-linker molecules. The exact stoichiometry, *r,* required to initiate these post-curing reactions depends on many factors, such as the type and molecular weight of the cross-linker used, as well as the concentration of the catalyst. Thus, in order to provide a guideline on how to develop stable and reliable silicone elastomers via condensation curing, a formulation optimization study was conducted. For hereby studied elastomer formulations, a stoichiometric ratio between 3 and 5 was found to produce the fewest post-curing reactions. Lower *r* values produced elastomers that become unstable over time due to the reaction between unreacted and/or dangling polymer chains. At higher *r* values, so-called cross-linker domains were formed via the reaction between redundant cross-linker molecules. In contrast to the condensation of unreacted and/or dangling polymer chains, this reaction occurred relatively quickly—within approximately 3 weeks—and therefore did not contribute to long-term elastomer instability.

The long-term stability study served as a tool for preparation of stable condensation curing silicone elastomers. The elastomer properties were then tailored by cross-linker domains density and concentration. The cross-linker domains density was altered by changing either cross-linker chain length (Di-10, Di-50, and Di-400) or cross-linker type ((MeO)_3_Si-PDMS-Si(OMe)_3_ and (QM^OEt^)_8_). The cross-linker domains concentration was altered by varying *r*. As expected, the Young´s modulus of the elastomers was increasing and elongation at break was decreasing with increasing *r.* The smallest change of these properties was observed when using the highest molecular weight cross-linker (Di-400), as the resulting cross-linker domains were less dense due to the long distance between cross-links. The presence of cross-linker domains was found to positively affect electrical breakdown strength, with the highest value obtained for E_C2T+(QM^OEt^)_8__r15 (130 µm/V). Scratch resistance was found to be highly dependent on both the size and rigidity/elasticity of the cross-linker domains. Due to their high weight percentage in the elastomer network, cross-linker domains Di-400 and Di-50 were found to have negligible and negative effects, respectively, on scratch resistance. On the other hand, scratch resistance was significantly improved by the presence of both Di-10 and (QM^OEt^)_8_ cross-linker domains. Scratch resistance comparable to that of the reference coating (E_Ref), which contained reinforcing fillers, was obtained for both E_C2T+Di-10_r15 and E_C2T+(QM^OEt^)_8__15. While E_C2T+Di-10_r15 performed well in both “single” and “multiple” scratch resistance tests, “multiple” scratch resistance was significantly lower for E_C2T+(QM^OEt^)_8__15, indicating the importance of not only cross-linker domains size, but also rigidity/elasticity.

## Figures and Tables

**Figure 1 molecules-26-00082-f001:**
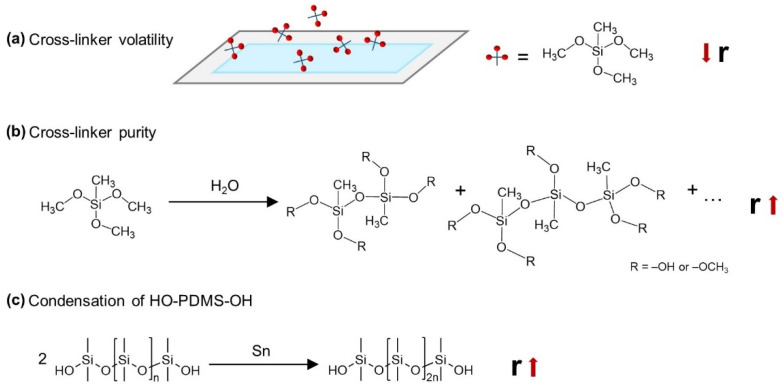
Factors influencing the true stoichiometric ratio, r = f[cross-linker]/2[HO-PDMS-OH], in a condensation curing silicone elastomer formulation [10]. The example shown is a system consisting of HO-PDMS-OH, methyltrimethoxysilane (f = 3), and a tin catalyst: (**a**) Cross-linker volatility, (**b**) hydrolysis-condensation reaction of the cross-linker molecules during storage leading to cross-linker impurity, and (**c**) condensation reaction of HO-PDMS-OH in the presence of a tin catalyst.

**Figure 2 molecules-26-00082-f002:**
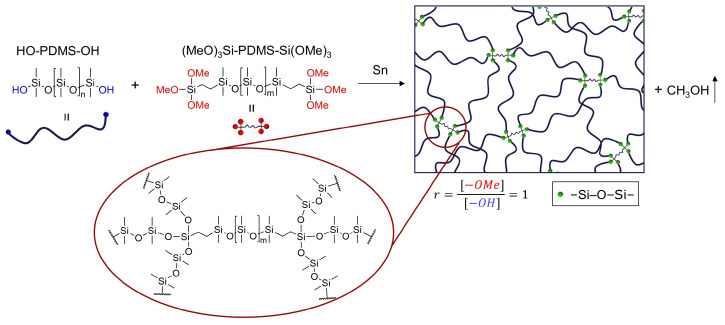
Example of an ideal network structure obtained by the reaction of HO-PDMS-OH with (MeO)_3_Si-PDMS-Si(OMe)_3_. The ideal network will be obtained if the components are used in a stoichiometric balance (*r* = 1), and no side reactions, such as condensation of HO-PDMS-OH and reaction between cross-linker molecules, are present. In addition, all functional groups will possess the same steric accessibility.

**Figure 3 molecules-26-00082-f003:**
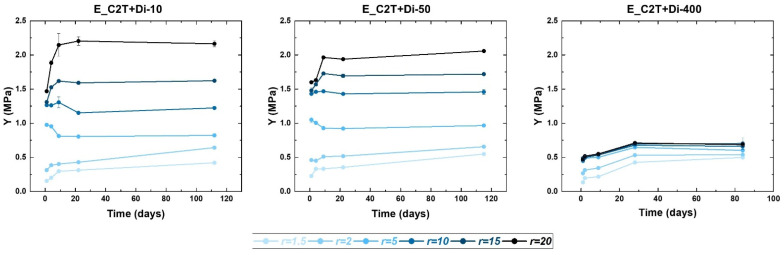
Stability of elastomer films assessed via change in Young´s modulus over time. The elastomers were prepared via the reaction between C2T and Di-10, Di-50, and Di-400, respectively. The stoichiometric ratio was varied from 1.5 to 20. Film thickness was ~100 µm.

**Figure 4 molecules-26-00082-f004:**
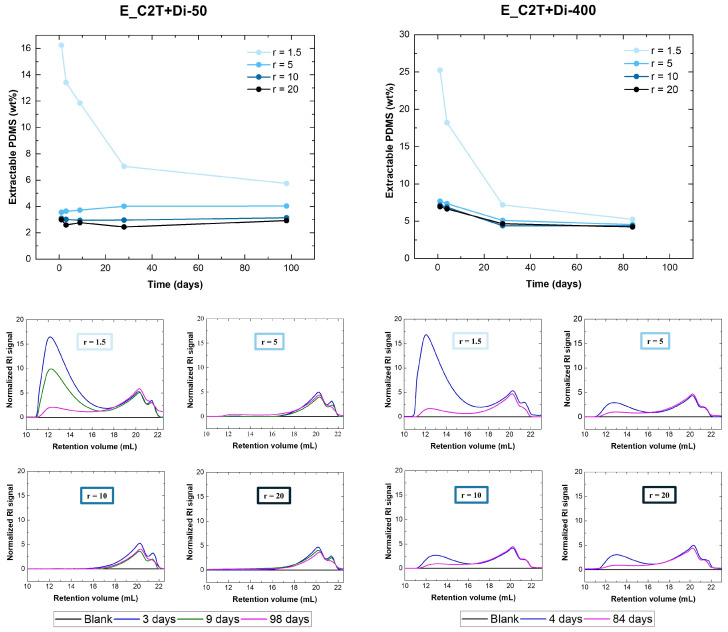
(**Top**) Extractable PDMS (wt%) as a function of time (determined from ^1^H NMR analysis). (**Bottom**) SEC eluograms of the extractable PDMS. Analyses were performed on elastomers prepared via the reaction of C2T with Di-50 and Di-400, respectively. The stoichiometric ratio was varied from 1.5 to 20. Film thickness was ~100 µm.

**Figure 5 molecules-26-00082-f005:**
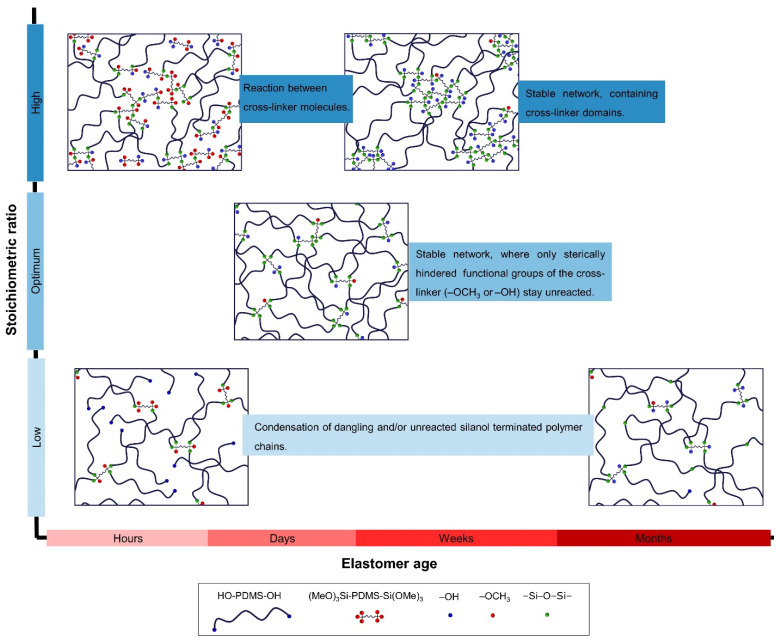
Schematic representation of the effects of stoichiometric ratio and elastomer age on elastomer network structure and stability.

**Figure 6 molecules-26-00082-f006:**
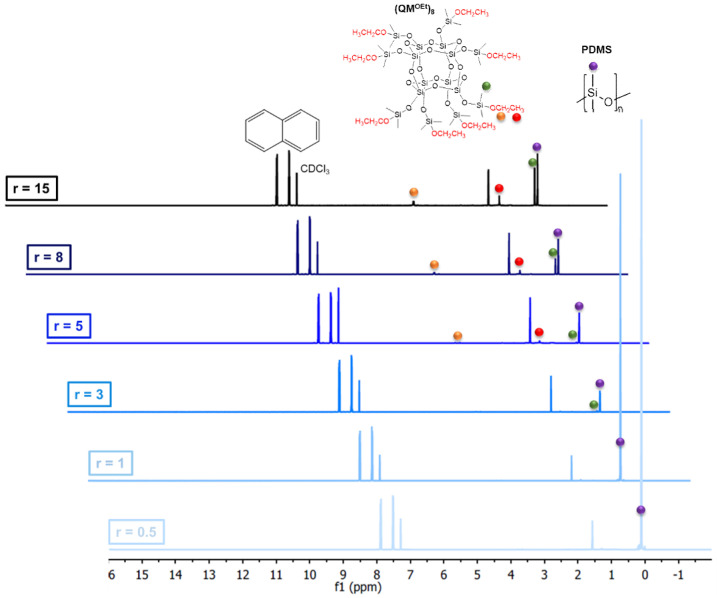
^1^H-NMR spectra of extracts from 1-day old elastomer films prepared via the reaction between C2T and (QM^OEt^)_8_. The stoichiometric ratio was varied from 0.5 to 15.

**Figure 7 molecules-26-00082-f007:**
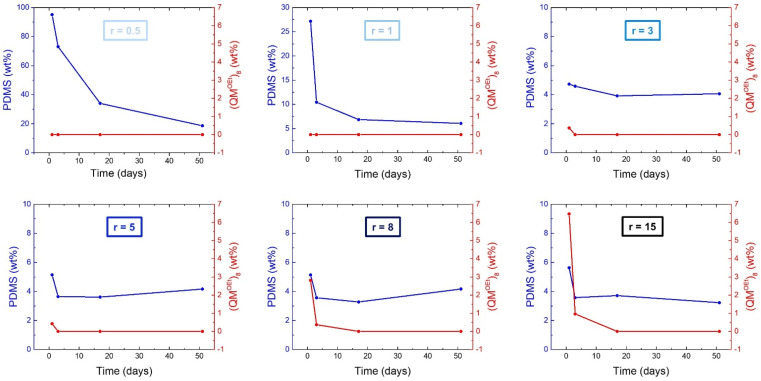
The amount of extractable PDMS and (QM^OEt^)_8_ as a function of time. Elastomers were prepared via the reaction between C2T and (QM^OEt^)_8_ using stoichiometric ratios of 0.5, 1, 3, 5, 8, and 15, respectively.

**Figure 8 molecules-26-00082-f008:**
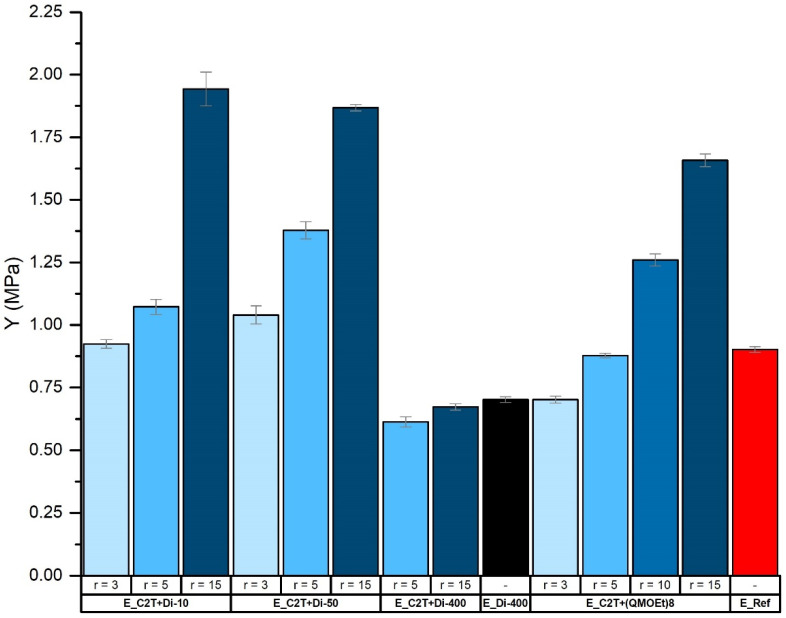
Young´s moduli (MPa) of commercial coating E_Ref and elastomers prepared via the reaction between C2T and Di-10, Di-50, Di-200, and (QMOEt)8, respectively. Film thickness was ~100 µm.

**Figure 9 molecules-26-00082-f009:**
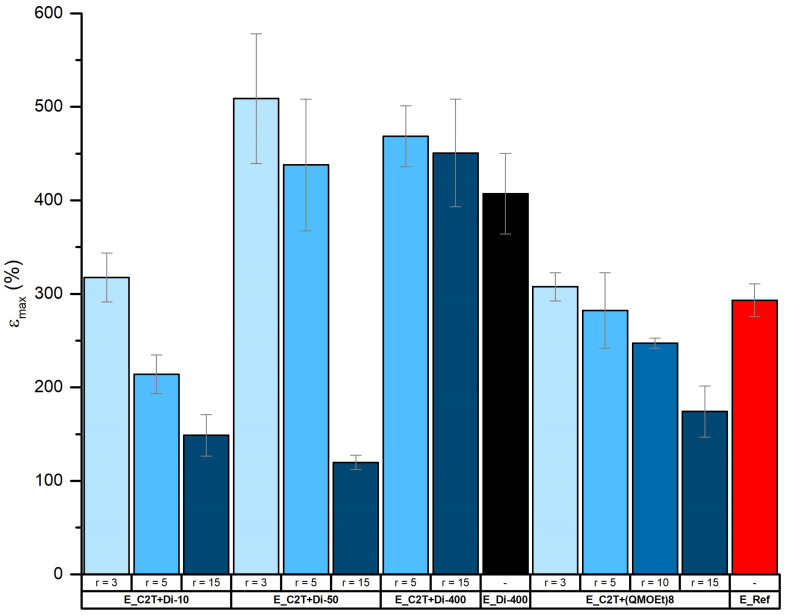
Elongation at break (%) of commercial coating E_Ref and elastomers prepared via the reaction between C2T and Di-10, Di-50, Di-200, and (QM^OEt^)_8_, respectively. Film thickness was ~100 µm.

**Figure 10 molecules-26-00082-f010:**
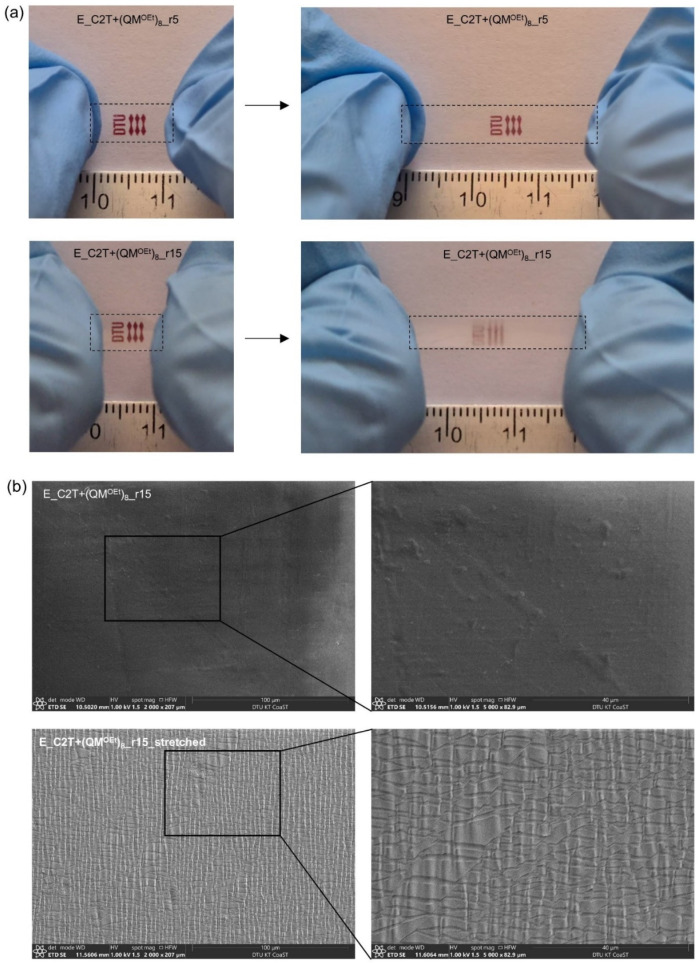
(**a**) Elastomers E_C2T+(QM^OEt^)_8__r5 and E_C2T+(QM^OEt^)_8__r15 before and after elongation. Elastomer samples are marked with a dashed line to improve visibility. The DTU logo is printed on the paper below the sample. (**b**) SEM images of E_C2T+(QM^OEt^)_8__r15 in unstretched and stretched state, respectively.

**Figure 11 molecules-26-00082-f011:**
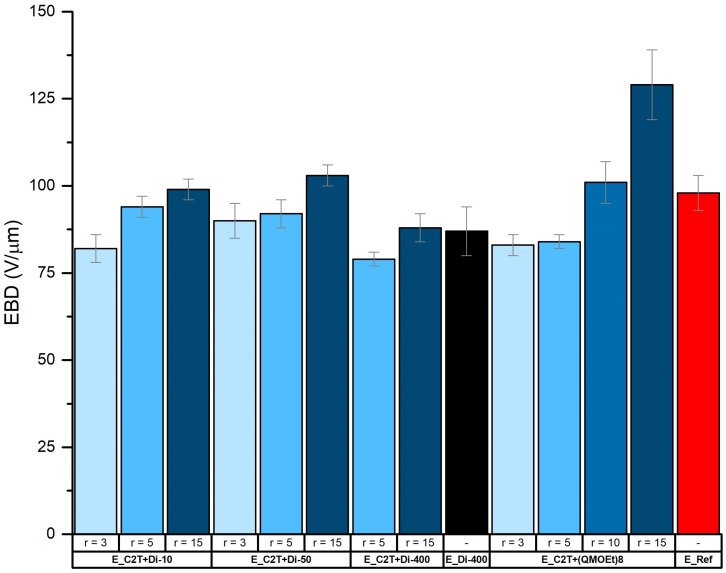
Electrical breakdown (EBD) strength (V/µm) of commercial coating E_Ref and elastomer films prepared via the reaction between C2T and Di-10, Di-50, Di-200, and (QM^OEt^)_8_, respectively. Film thickness was ~100 µm.

**Figure 12 molecules-26-00082-f012:**
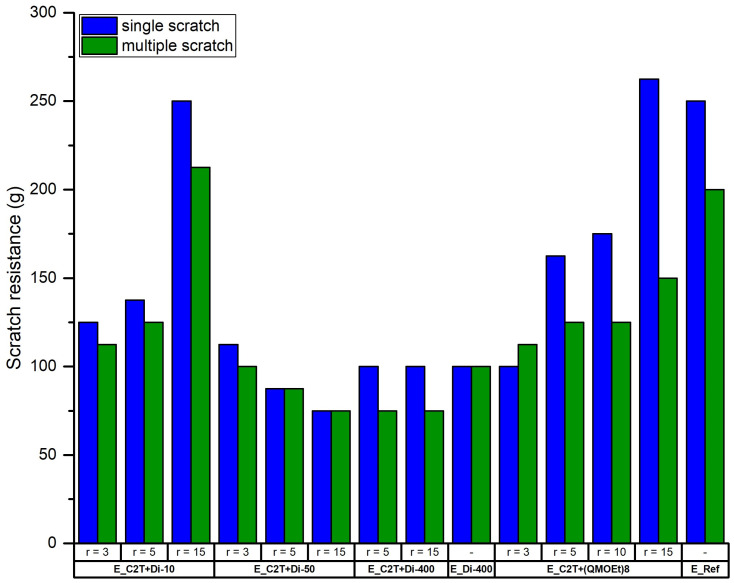
Scratch resistance of commercial coating E_Ref and elastomer films prepared via the reaction between C2T and Di-10, Di-50, Di-200, and (QM^OEt^)_8_, respectively. Film thickness was ~100 µm.

**Figure 13 molecules-26-00082-f013:**
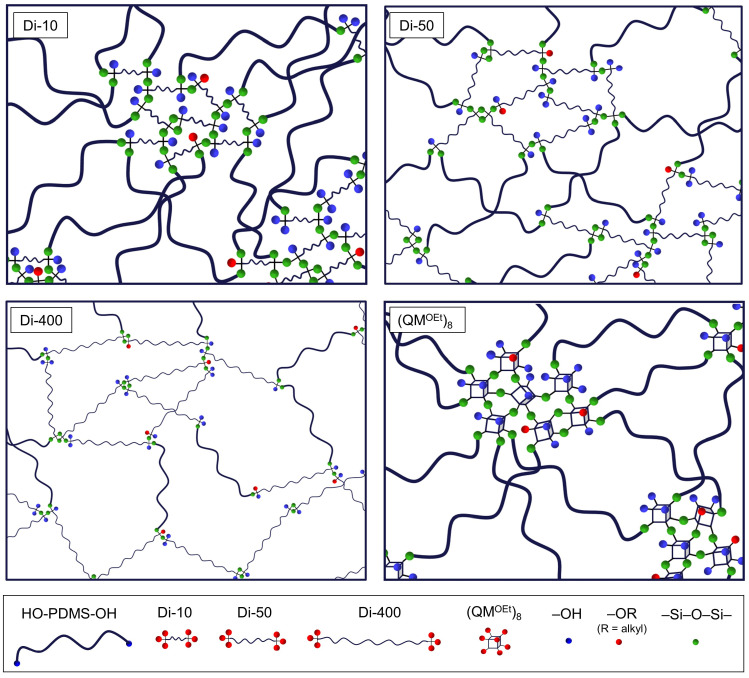
Simplified illustration of network structures prepared at high *r* using Di-10, Di-50, Di-400, and (QM^OEt^)_8_ cross-linker, respectively.

**Figure 14 molecules-26-00082-f014:**
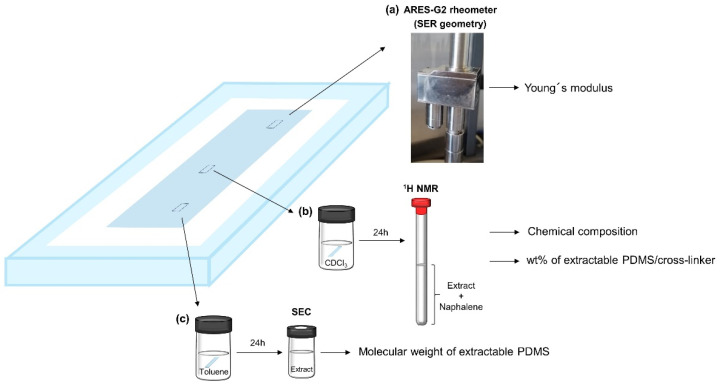
Methods used to evaluate the stability and network structure of elastomer films: (**a**) ARES-G2 rheometer equipped with SER geometry was used to track changes in Young´s modulus over time;(**b**) ^1^H NMR was used to analyze the chemical composition of the extract and to determine the wt% of extractable PDMS/cross-linker; (**c**) SEC was used to measure the molecular weight of extractable PDMS.

**Table 1 molecules-26-00082-t001:** Summary of compounds used in the elastomer formulations.

Compound	Abbreviation	Molecular Weight (g/mol)	Chemical Structure
HO-PDMS-OH	C2T	20,200 *	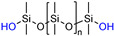
(MeO)_3_Si-PDMS-Si(OMe)_3_	Di-10	1200 *	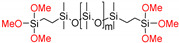
	Di-50	5500 *	
	Di-400	20,300 *	
(QM^OEt^)_8_	(QM^OEt^)_8_	1370.4 **	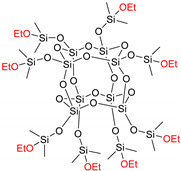

* Number average molecular weight (M_n_) determined by SEC; ** determined from molecular structure.

## Data Availability

The data can be obtained from the corresponding author.

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
