# Peer review of "Reliable Condensation Curing Silicone Elastomers with Tailorable Properties"

_molecules, 2020, doi:10.3390/molecules26010082_

Round 1

Reviewer 1 Report

I

In this manuscript, the authors try to make a fine control of the properties of condensation curing silicones for use as thin, reliable coatings. Two ways of control are approached: elastomer formulation and post-curing.

Regarding the first aspect, polydimethylsiloxane-a,w-diol is crosslinked with relatively high molecular weight agents with low volatility, able to ensure the preservation of stoichiometry regardless of conditions and sample size, namely trimethoxysilane-terminated polysiloxane ((MeO)3-PDMS-(OMe)3) with different chain lengths and ethoxy-terminated octakis(dimethylsiloxy)-T8-silsesquioxane.

For this, the researches were carried out according to an experimental program in which the stoichiometry of the crosslinking system is varied and the variation in time of the Young modulus is followed. Moreover, soluble fractions are extracted and analyzed by SEC and NMR.

The performances of the obtained films were evaluated by investigating the morphology, the mechanical properties (Young modulus, elongation, scratch resistance) and the electrical strength, the latter as an indicator of the homogeneity of the films.

It was found that the properties of the resulting elastomer can be tuned by the type of crosslinker, its concentration and molecular weight. It was also found that both the stability of the mechanical properties over time and the final properties of the resulting elastomers are significantly affected by two post-curing reactions, namely the condensation of dangling and/or unreacted polymer chains and the reaction between crosslinker molecules.

The research is well conducted, the results are correctly interpreted and useful for controlling the properties of the resulted silicones.

However, it would be good if several arguments were presented to support the usefulness of these silicones as coatings. What other properties besides scratch resistance do recommend them for this? What can be said about adhesion to the substrate, for example?

Author Response

Dear Editor, dear reviewers,

Thank you for the constructive comments raised by both reviewers. We have modified the manuscript following the reviewers´ suggestions. Below, we have addressed each issue raised and also uploaded a file showing ‘track changes’. Thank you for considering this manuscript. 

Best regards

Alena Juraskova

(on behalf of the authors)

Reviewer 1

  1. However, it would be good if several arguments were presented to support the usefulness of these silicones as coatings. What other properties besides scratch resistance do recommend them for this? What can be said about adhesion to the substrate, for example?

Answer: Thank you for this comment. Following paragraph has been added (line 268-273): “Scratch resistance, together with coating/substrate adhesion, is one of the most important parameters for materials used as protective coatings. A poor scratch resistance and/or adhesion to the substrate can lead to reduction of coating performance, reliability, and lifetime.[1, 30-32] In this study, all the tested elastomers, independently of their composition, showed initial cohesive failure before failing adhesively when scratched, signifying a good adhesion to their substrate, namely Hempel's Nexus II 27400.[30, 31]”

Reviewer 2 Report

Title: Reliable condensation curing silicone elastomers with tailorable properties. (Submitted to Molecules)

   In this paper, the long-term stability of the condensation-curable silicone elastomer and the properties according to the curing conditions are systematically evaluated. This typed study is very fundamental and important from a industrial viewpoint. Generally, the interpretation of the data draws reasonable conclusion. This manuscript is clearly written with all necessary results and discussions. However, the authors would need to address several issues before publication.

Basically, I recommend the publication of the manuscript by Molecules, regarding its important information to the field of polymer material.

・It is better to describe the abbreviation of trimethoxysilane-terminated polydimethylsiloxane as (MeO)3Si-PDMS-Si(OMe)3.

・Although the reviewer almost agree with the schematic illustration as seen in Figure 5 and 13, can the domain size shown in the Figure be actually measured by AFM or other measurements?

・The reviewer cannot understand why the opacity seen in Figure 10 occures. Is such a phenomenon observed even when Di-10 is used? If it is observed when using cubic molecules, is it probably because stretching caused the molecules to aggregate and/or crystallize?

・In table A2, the sample name of C2T+Di-400 is incorrect.

Author Response

Dear Editor, dear reviewers,

Thank you for the constructive comments raised by both reviewers. We have modified the manuscript following the reviewers´ suggestions. Below, we have addressed each issue raised and also uploaded a file showing ‘track changes’. Thank you for considering this manuscript. 

Best regards

Alena Juraskova

(on behalf of the authors)

Reviewer 2

  1. It is better to describe the abbreviation of trimethoxysilane-terminated polydimethylsiloxane as (MeO)3Si-PDMS-Si(OMe)3.

Answer: The abbreviation has been changed throughout the manuscript. Specifically, the changes have been made in lines 21, 71, 111, 174, 321, 322, 338, 436, Figure 2, Figure 5, Table 1, and in the description of Figure A1 (line 472).

  1. Although the reviewer almost agree with the schematic illustration as seen in Figure 5 and 13, can the domain size shown in the Figure be actually measured by AFM or other measurements?

Answer: Thank you for the question. We agree that AFM would be a good support of the schematic illustration in Figure 5 and 13. Nevertheless, due to Covid-19 restrictions, we have not been able to conduct AFM analysis of our silicone films. Despite the missing AFM images, we believe that the rheology, SEC and NMR data presented in the manuscript are enough to support the illustrations, which are only schematics with no exact domains sizes, as AFM or other measurements would be needed to obtain this information.

  1. The reviewer cannot understand why the opacity seen in Figure 10 occurs. Is such a phenomenon observed even when Di-10 is used? If it is observed when using cubic molecules, is it probably because stretching caused the molecules to aggregate and/or crystallize?

Answer: Yes, the opacity is observed only for the E_C2T+(QMOEt)8_r15. From the SEM images (Figure 10) can be seen that the surface of the elastomer changes significantly upon stretch. As the reviewer indicates, we also believe that the observed surface change is a result of the cubic cross-linker domains aggregation and crystallization.

For clarity, the following paragraph has been added (line 235-241): “Additionally, the elastomer E_C2T+(QMOEt)8_r15 exhibited opacity when elongated. For better understanding of this behavior, which was not observed for any of the other elastomers, the elastomer E_C2T+(QMOEt)8_r15 was investigated using SEM in both unstretched and stretched state. The sample preparation procedure can be found in Figure A5. As it can be seen in Figure 10, the elastomer undergoes a significant surface change when stretched, which explains the observed opacity and is believed to be a result of (QMOEt)8 domains aggregation and crystallization.”

  1. In table A2, the sample name of C2T+Di-400 is incorrect.

Answer: Thank you for catching this. The samples names in Table A2 have been corrected.